# Vitamin D Intake and Serum Levels in Pregnant and Postpartum Women

**DOI:** 10.3390/nu15153493

**Published:** 2023-08-07

**Authors:** Laila Meija, Anna Piskurjova, Ksenija Nikolajeva, Olga Aizbalte, Roberta Rezgale, Aivars Lejnieks, Vinita Cauce

**Affiliations:** 1Department of Sports and Nutrition, Rīga Stradiņš University, 26a Anniņmuižas bulvāris, LV-1067 Riga, Latvia; 2Riga East Clinical University Hospital, 2 Hipokrata Street, LV-1038 Riga, Latvia; ksenija.nikolajeva@rsu.lv (K.N.); aivars.lejnieks@rsu.lv (A.L.); 3Department of Public Health and Epidemiology, Rīga Stradiņš University, 9 Kronvalda bulvāris, LV-1010 Riga, Latvia; anna.piskurjova@rsu.lv (A.P.); olga.aizbalte@rsu.lv (O.A.); vinita.cauce@rsu.lv (V.C.); 4Doctoral Department, Faculty of Medicine, Rīga Stradiņš University, 16 Dzirciema Street, LV-1007 Riga, Latvia; 5Faculty of Medicine, Rīga Stradiņš University, 16 Dzirciema Street, LV-1007 Riga, Latvia; 023609@rsu.edu.lv; 6Department of Internal Diseases, Rīga Stradiņš University, 16 Dzirciema Street, LV-1007 Riga, Latvia

**Keywords:** pregnancy, vitamin D, supplementation, dietary intake

## Abstract

Maternal vitamin D deficiency, which is highly prevalent in pregnant women in Europe, is linked to adverse health effects for both the mother and child. The objective was to assess vitamin D status in pregnant women by evaluating their dietary and supplemental vitamin D intake, serum vitamin D levels, parathyroid hormone levels, and lifestyle factors. This cross-sectional study, with a total of 735 participants (145 pregnant and 590 up to the seventh day postpartum), took place in Latvia. Blood samples, a food frequency questionnaire, and medical documentation were used for data collection. The median serum vitamin D concentration was 34.0 ng/mL, with pregnant women having higher levels (42.9 ng/mL) than postpartum women (31.8 ng/mL). There was no association between vitamin D serum concentration and dietary intake of vitamin D (*p* > 0.05), whereas there was a significant correlation with use of vitamin D supplements (r = 0.41; *p* < 0.001 in pregnant women and r = 0.35; *p* < 0.001 in postpartum women). This study demonstrated that a minority of pregnant women (21.9%) had optimal serum vitamin D concentration (>45 ng/mL), and diet had no significant impact on vitamin D levels. Thus, our proposed recommendation for vitamin D intake during pregnancy was 63 mcg (2500 IU) year-round for optimal levels in pregnant women in Northeastern Europe.

## 1. Introduction

Vitamin D has broad therapeutic potential during pregnancy for both the mother and the fetus, aside from its usual role in the endocrine regulation of calcium and bone metabolism. Low vitamin D levels can adversely affect the course and outcome of pregnancy [1,2]. Maternal vitamin D deficiency during pregnancy has been associated with risk of pre-eclampsia, preterm birth, insulin resistance, bacterial vaginosis, gestational diabetes, osteomalacia, and muscle weakness, as well as an increased risk of immune dysfunction in the mother [3,4]. In neonates, adverse outcomes such as small gestational age, preterm birth, smaller size at birth, a detrimental effect on bone and teeth development, increased risk of infectious disease, and an influence on central nervous system development were reported [5].

It is accepted that determining serum level 25-hydroxyvitamin [25(OH)D] serves as the best method for evaluating vitamin D status [5,6]. To date, there is no global consensus on reference values for 25(OH)D levels used to identify a vitamin D deficiency and insufficiency in pregnancy [7,8]. Sufficiency is defined by different serum levels of 25(OH)D: 20–50 ng/mL by I U.S. Institute of Medicine (IOM) [9], and 30 ng/mL by the U.S. Endocrine Society [10], whereas other studies proposed threshold reference values of 40 ng/mL [11,12] or 50 ng/mL to provide a broader spectrum of benefits, especially for the immune system [13,14].

Maternal vitamin D deficiency is common during pregnancy across the globe, especially among women with limited sun exposure who live in cold northern latitudes or wear sun-protective clothing [1,15]. Data show that, on average, half of pregnant and lactating women in Europe have vitamin D insufficiency (25(OH)D < 30 ng/mL) and one-tenth of pregnant women have vitamin D deficiency (defined as 25(OH)D < 20 ng/mL) [16].

The appropriate intake of vitamin D from food and/or supplements, as well as sun exposure for vitamin D synthesis in the skin, is needed to maintain optimal 25(OH)D serum concentrations [17]. Obesity [18,19,20], low physical activity levels [21], and genetic factors [17] play an important role in the response of 25(OH)D serum levels to vitamin D supplementation during pregnancy.

Vitamin D concentration is closely related to another calcium homeostasis regulator: parathyroid hormone (PTH) concentration. When vitamin D concentration decreases, so does calcium absorption, which induces an increase in PTH secretion. This affects many processes, including the intestinal absorption of vitamin D and calcium, conversion of vitamin D in the skin, reabsorption of calcium from urine, and resorption from bones [6,22,23]. PTH concentration has been considered an indicator of vitamin D deficiency [11]. As the optimal levels of PTH and vitamin D in pregnancy are not well established, the clinically relevant 25(OH)D threshold at which PTH is suppressed is still under debate [23].

The aim of this study was to investigate vitamin D status in pregnant women by evaluating their vitamin D intake from food and supplements, serum vitamin D levels, PTH levels, and lifestyle factors.

## 2. Materials and Methods

### 2.1. Study Design and Population

This was a cross-sectional study with a total of 735 participants. The study took place in outpatient clinics and maternity wards in all Latvian regions from July 2020 to January 2023, including all seasons.

The inclusion criteria: pregnant women in the 27th to 40th week of gestation; adult and postpartum women up to the 7th day after delivery (postpartum women) who had resided in Latvia for more than one year.

The exclusion criteria: age under 18, women without legal capacity, multifetal pregnancy, previously diagnosed metabolic diseases: diabetes mellitus, celiac disease, short bowel syndrome, Crohn’s disease, ulcerative colitis, eating disorders, and history of gastrointestinal surgery, HIV/AIDS infection.

### 2.2. Data Collection

For data collection, a food frequency questionnaire about the participants’ diet during the last six months (frequency and quantity of 211 food products) was used. For the present study, the questionnaire was adapted from the one developed by the Scientific Institute of Food Safety, Animal Health and Environment “BIOR” (BIOR) for evaluating nutritional data on the population of Latvia [24]. The questionnaire contained 199 products (including consumption of vitamin-D-containing food such as fish, dairy products, eggs, mushrooms) and beverages, which were grouped into 20 product groups as well as supplements (including multivitamins and vitamin D supplements). For evaluation of the portion size of foods, the “Photo Atlas of Food Products and Food Portions” was used [24]. The interview was performed by trained interviewers (registered nutritionists, medical students, and residents).

A questionnaire on demographic data (age, education, marital status, occupation), dietary habits, and lifestyle, such as physical activity during pregnancy and smoking, was also administered.

In addition, medical data, data on pregnancy complications (labor induction, anemia, fetal macrosomia, gestational diabetes, preterm delivery, urinary tract infection, pre-eclampsia), weight (kg) and height (cm) of the women, results of blood tests, and anthropometric parameters of the newborn at birth were collected from medical documentation.

Body mass index (BMI) (kg/m^2^) was classified into underweight (<18.5 kg/m^2^), normal weight (18.5–24.9 kg/m^2^), overweight (25.0–29.9 kg/m^2^), and obese (>30.0 kg/m^2^) according to the WHO criteria [25].

### 2.3. Blood Samples and Cut-Off Points for Vitamin D and PTH Levels

Blood samples were collected from the antecubital vein. The blood test was non-fasting and the sample was collected by a laboratory nurse. The serum 25(OH)D concentration was determined using chemiluminescent microparticle immunoassay (CMIA) method on the Alinity System, ABBOTT.

For data analysis, the latest data on blood vitamin D reference range were taken into account. Women were classified into groups defined by their vitamin D status. Vitamin D was considered critically low when serum 25(OH)D ≤ 10 ng/mL, insufficient when 10–29.9 ng/mL, normal when 30–44.9 ng/mL, optimal when 45–54.9 ng/mL, and upper range of optimal was >55 ng/mL [10,26].

In addition, women were divided into two groups: with serum 25(OH)D concentrations below 45 ng/mL and above 45 ng/mL.

Serum parathyroid hormone (PTH) was determined using chemiluminescent microparticle immunoassay (CMIA) on the Alinity System, ABBOTT, with reference limits for adults (1.48–7.63 pmol/L).

### 2.4. Ethical Considerations

The study was approved by the Clinical Research Ethics Committee of Riga Stradiņš University (N.6-1/02/62). The study participants were provided with written information on the study’s aim and the objective of the study and all participants provided written consent to their participation in the study. The study was conducted anonymously and the obtained data were encrypted with a code assigned to each participant.

### 2.5. Data Analysis

Dietary data from the dietary frequency questionnaire were processed at BIOR using a program based on Microsoft Dynamics Ax 2009 developed by the institute. The program uses the Latvian Food Composition Database, which was first made for the “Food Consumption Study of the Latvian Population (2012–2013)” and is still used to analyze food consumption in other studies on the Latvian population.

Descriptive statistical methods were used for data analysis: quantitative variables are described by a median with interquartile range (from Q1 to Q3), minimum, and maximum. Categorical data are described by count and percentage. The consistency of the quantitative variables with a normal distribution and the homogeneity of the data were determined using graphical methods and Shapiro–Wilk and Levene tests.

To verify the significant difference in quantitative variables between groups, the non-parametric Mann–Whitney or the Kruskal–Wallis H test with Bonferroni correction were used. The chi square test was used to compare categorical values. Binary logistic regression was used adjusting OR and 95%CI for characteristics of the study population. The relationship between two variables was assessed using Spearman’s correlation coefficient. A *p*-value < 0.05 was considered statistically significant. The data were processed using Microsoft Excel and SPSS version 27.0.

## 3. Results

### 3.1. General Characteristics

The general characteristics of participants are shown in Table 1. The median (Mnd) age of all participants was 31 (from 28 to 36) years; 145 (19.7%) were pregnant women and 590 (80.3%) were postpartum women. Median BMI was 22.4 (20.5 to 25.3) kg/m^2^, ranging from 15.2 to 44.5 kg/m^2^. Pregnant women had a lower pre-pregnancy BMI, higher prevalence of higher education and marriage, and longer walking/cycling time per day than postpartum women.

### 3.2. Dietary Intake

Daily energy and macronutrient intake is shown in Table 2. The median of energy intake from macronutrients for both groups was accounted for as follows: 18% protein, 38% carbohydrates, and 42% fat.

Vitamin D supplements (without taking into account multivitamin supplements) were taken during the last six months by 67.5% (*n* = 387) of postpartum women and by 84.1% (*n* = 122) of pregnant women. The intake of vitamin D from food and supplements in micrograms (mcg) was analyzed and data are represented in Table 3.

The main food sources for the consumption of vitamin D were assessed (g per week). The results are shown in Table 4.

### 3.3. Serum Vitamin D and PTH

The median serum vitamin D concentration (*n* = 735) was 34.0 (from 23.6 to 43.1) ng/mL (min 8.0, max 92.5 ng/mL) and PTH concentration (*n* = 725) was 3.0 (from 2.1 to 4.3) pmol/L (min 8.0, max 12.4 pmol/L) in pregnant and postpartum women taken together.

The median serum vitamin D concentration in pregnant women was 42.9 (from 32.7 to 51.8), which was higher than the rates reported in postpartum women, at 31.8 (from 21.8 to 40.0) ng/mL (*p* < 0.001).

The median PTH concentration in pregnant women was 2.8 (from 2.1 to 4.1), and in postpartum women, it was 3.0 (from 2.1 to 4.3) pmol/L (*p* = 0.397).

Figure 1 shows the distribution of different vitamin D serum concentration thresholds in pregnant and postpartum women.

Dividing participants in two groups with serum levels of vitamin D of <45 ng/mL and >45 ng/mL showed that the proportion with higher vitamin D levels was higher in pregnant women 64 (44.1%) than in postpartum women 97 (16.4%) (*p* < 0.001), with OR 3.2 (95%CI 2.1 to 4.9) adjusted by BMI group, education level, marital status, and walking/cycling time per day. Only 161 (21.9%) of all women had a vitamin D serum concentration of >45 (ng/mL).

There was a difference between both groups, pregnant and postpartum women, in vitamin D serum levels <45 ng/mL and >45 ng/mL and vitamin D supplement use (*p* = 0.018, respectively *p* < 0.001). All pregnant women with serum vitamin D level >45 ng/mL used vitamin D supplements.

There were no differences between age, BMI, dietary intake of kcal, protein, carbohydrate, fat, vitamin D dietary intake, and use of multivitamins in pregnant and postpartum women (*p* > 0.05).

In the group with serum vitamin D levels of >45 ng/mL, participants smoked less (*p* = 0.034) and were more educated (*p* = 0.03).

The median vitamin D intake due to vitamin D supplements in all participants with vitamin D serum levels of >45 ng/mL was 62.5 (25.0 to 112.5) mcg/d and in participants with vitamin D serum levels of <45 ng/mL was 62.5 (25.0 to 62.5) mcg/d (*p* < 0.001).

There was a significant difference (*p* < 0.001) in median vitamin D serum levels in women who took vitamin D supplements and who did not take supplements for both pregnant 54.4 (from 49.1 to 70.0) ng/mL vs. 33.0 (from 27.6 to 39.6) ng/mL) and postpartum women 52.4 (from 48.1 to 57.8) ng/mL vs. 28.7 (from 19.8 to 35.4 ng/mL).

Women who used vitamin D supplements had a higher educational level (*p* < 0.001) and were non-smokers (*p* < 0.001), but showed no difference from women who did not use supplements in parameters such as BMI, age, and family status.

The seasonal vitamin D serum concentration is represented in Figure 2. In all seasons, serum vitamin D concentration was higher in pregnant than in postpartum women (*p* < 0.05). There were no statistically significant differences in vitamin D serum levels between seasons in pregnant or in postpartum women (*p* > 0.05). The only difference was found in pregnant women without vitamin D supplementation: their vitamin D serum concentration was significantly higher in summertime than in winter and spring seasons (*p* = 0.006 vs. *p* = 0.011).

There was no association between vitamin D serum concentration and dietary intake of vitamin D, either in overall vitamin D dietary intake or separately with intake of fish, milk products, mushrooms, and eggs (*p* > 0.05), whereas there was a significant correlation with the use of vitamin D supplements (r = 0.41; *p* < 0.001 in pregnant women and r = 0.35; *p* < 0.001 in postpartum women) but no association between vitamin D serum concentration and the intake of multivitamin supplements containing vitamin D for pregnant women (*p* > 0.05). In both pregnant and postpartum groups, supplement users obtained a median amount of vitamin D with supplements 62.5 (from 25.5 to 112.5) mcg.

Vitamin D serum concentration was associated with the number of births in pregnant women and postpartum women, and only in postpartum women with PTH levels. In all participants, associations between serum vitamin D level and BMI, walking/cycling time per day and education were found, whereas no associations were found with age, time since previous delivery, or gestational weeks at delivery (*p* > 0.05).

There was an association between vitamin D serum levels in postpartum women and birth weight (r = −0.09; *p* = 0.031). The associations between vitamin D serum levels and influencing factors are presented in Table 5.

PTH concentration in pregnant women was only associated with age (r = 0.18; *p* = 0.036), while in postpartum women, it was associated with vitamin D serum concentration (r = −0.21; *p* < 0.001), BMI (r = 0.11; *p* = 0.012), time since previous delivery (r = −0.11; *p* = 0.037), and time of walking/cycling per day (r = −0.09; *p* = 0.038). No associations were found with number of births, birth weight, intake of vitamin D by food, multivitamins, or vitamin D supplements (*p* > 0.05).

## 4. Discussion

The main concern of any clinician that has to face the dietary and supplementation recommendations on pregnant women is which guidelines, from all the inconsistency existing, should they apply. The present Latvian study aimed to evaluate vitamin D status of pregnant women, considering factors such as dietary and supplemental vitamin D intake, serum levels, PTH levels, and lifestyle habits. In our study, we found a median serum vitamin D concentration of 34.0 ng/mL, with only a minority of participants (21%) having optimal serum vitamin D concentrations (>45 ng/mL). The dietary intake of vitamin D for pregnant and postpartum women was found to be 2.5 mcg/d (100 IU) and 2.0 mcg/d (80 IU), respectively. Notably, vitamin D supplementation was the key factor in maintaining optimal vitamin D levels. The intake of vitamin D by supplements was 62.5 mcg for both groups.

The results of our study highlight that the median serum 25(OH)D found in our study was slightly higher than that found in European studies of pregnant women (28.0 ng/mL) and significantly higher than the median found in the Swedish study (19 ng/mL), respectively [27]. This difference can be likely explained by the increased interest in vitamin D supplementation during the COVID-19 pandemic during which our study was conducted due to some evidence that vitamin D may have a protective role against infection and complications in pregnant women [28,29].

Vitamin D deficiency is widespread in pregnant women worldwide. Different national guidelines set optimal serum vitamin D levels at around 20–50 ng/mL and offer different recommendations for supplementation, with mentioned daily doses ranging from 400 IU to 4000 IU [30,31]. In addition, there are many different interpretations and conflicting evidence on the possible effects of vitamin D deficiencies or overdoses on maternal and fetal health. Blood tests are used to determine vitamin D levels, allowing for individual adjustment of the vitamin intake. However, this is not always systematically applied and recommended by medical professionals.

The main dietary sources of vitamin D for the study respondents were fish, eggs, and dairy products. Getting enough vitamin D from food alone is commonly acknowledged to be difficult. In a study in Canada, Aghajafari et al. concluded that pregnant women have a low dietary intake of vitamin D from foods of 5 mcg (200 IU/d) [32], which is consistent with the findings of a study by Larquéa et al. [1] and is supported by our study findings as well, determining why dietary vitamin D intake alone does not provide optimal levels of vitamin D in serum.

The average dietary energy intake of our study population was within the recommended range, but the nutrient distribution did not match the recommended pregnancy distribution [33], with protein accounting for 10–20%, fat for 30%, and carbohydrates for 45–60% of the total energy. The study participants consumed less protein and carbohydrates than recommended, while fat intake exceeded recommended levels by 10%. However, neither dietary energy intake nor nutrient distribution influenced vitamin D levels.

A large proportion of our study participants took vitamin D supplementation during pregnancy. Serum vitamin D concentration in pregnant women was higher (42.9 ng/mL) than in postpartum women (31.8 ng/mL). This is explained by the finding that more pregnant women (84.1%) used vitamin D supplements than postpartum women (67.5%) used vitamin D, which is associated with the higher education of the pregnant women. However, the interest of this study is only on pregnancy, for which the half-life of 15 days of 25(OH)D should be considered, as this means that the results reflect vitamin D levels at the end of the pregnancy.

Despite the finding that the median vitamin D serum level of pregnant women in Latvia was higher than that found in European studies of pregnant women and the number of participants who took supplements was high, more than one third of our study participants *n* = 302 (41.1%) had vitamin D deficiency or insufficiency according to the US recommendation (<30 ng/mL) [10].

The World Health Organization recommends a dietary intake with supplementation, starting with 200 IU/day [34]. According to the Latvian Ministry of Health, the recommended intake of vitamin D during pregnancy is 400 IU vitamin D daily. In practice, higher doses are recommended by medical professionals based on recent research in other countries, which claim that taking more vitamin D (1000–4000 IU/day) could keep serum levels optimal during pregnancy [31,35]. Our research supports this observation, as the overall dietary intake from supplements in pregnant women appeared to be 2500 IU/d.

Our study data were collected throughout the year and there was no statistically significant seasonal difference in vitamin D serum levels. However, those women who did not take any vitamin D supplementation had higher levels of serum vitamin D in the summer season, which is most likely due to the increased sun exposure. This finding is also in line with other studies [36,37].

Various scientific studies reported different 25(OH)D cut-off points, ranging from 8.0 ng/mL to 30 ng/mL [38,39], at which the PTH starts to rise, and potentially threaten the resorption of calcium from the bones. However, in our study, PTH concentration in both groups was in normal range (1.48–7.63 pmol/L). Therefore, this cannot be determined in the work at hand.

During the revision of our data, we found that the increased interest in vitamin D supplementation during the COVID-19 pandemic for all the populations might be a limitation for our study. Even though the vitamin D serum samples were taken between the third trimester and seventh day postpartum, all of the samples reflected pregnancy due to the midlife of 25(OH)D. In spite of these limitations, there was a large number of participants and year-round comprehensive information on their food and supplements.

## 5. Conclusions

In this study, we found a considerable prevalence of vitamin D insufficiency in pregnant and postpartum women, with only a minority (21.9%) having optimal serum vitamin D concentrations (>45 ng/mL). Neither diet nor the use of multivitamins affected vitamin D levels. Only vitamin D supplementation played a role in maintaining optimal vitamin D levels. Users of vitamin D supplements were more educated and had lower rates of smoking. The seasons of the year did not influence vitamin D levels, except for women who did not take vitamin D supplements, as they had higher vitamin D concentrations in the summertime.

The proposed recommendations for vitamin D intake during pregnancy could consider a dose of 63 mcg (2500 IU) all year round, which should be appropriate for maintaining optimal vitamin D levels in pregnant women in Northeastern Europe. However, this needs to be re-evaluated in future studies.

## Figures and Tables

**Figure 1 nutrients-15-03493-f001:**
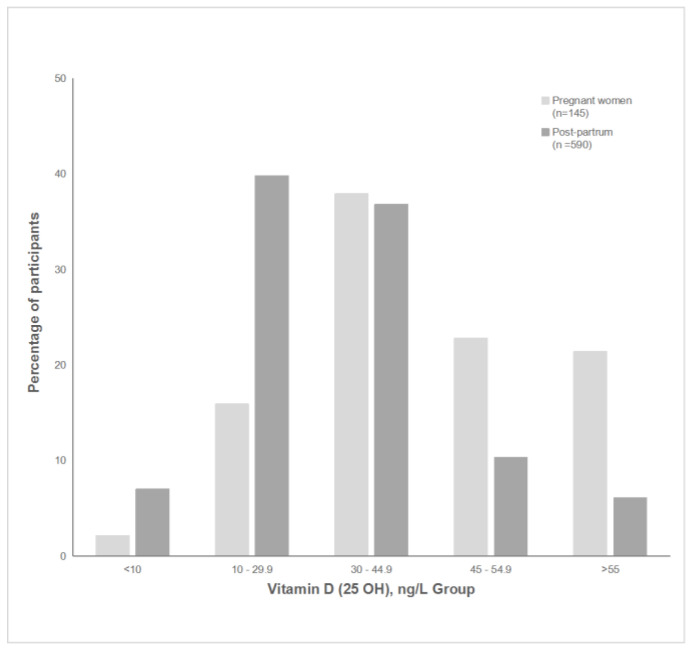
Vitamin D serum concentration by thresholds.

**Figure 2 nutrients-15-03493-f002:**
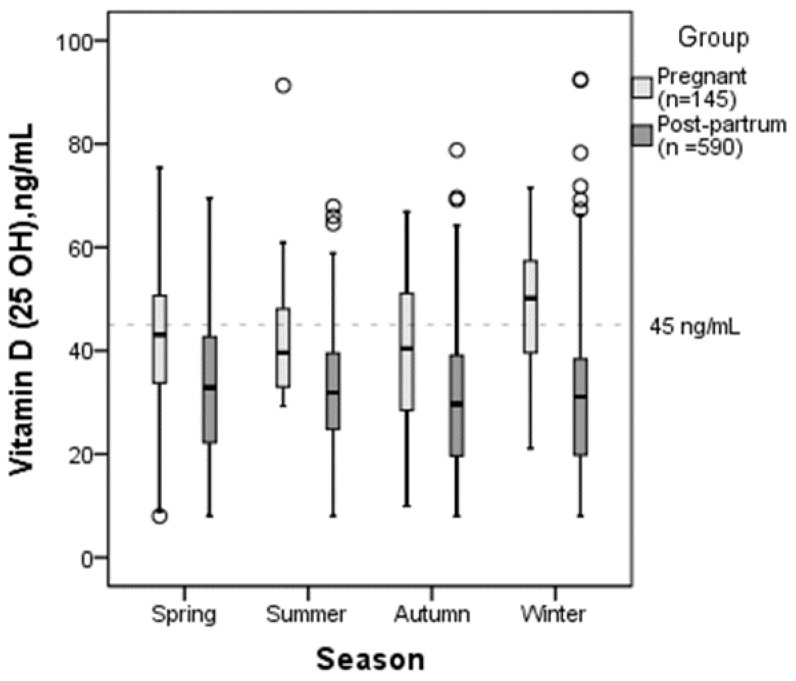
Vitamin D concentration in pregnant and postpartum women in different seasons.

**Table 1 nutrients-15-03493-t001:** General characteristics of the study population.

Characteristics	Pregnant		Postpartum		*p*
	(*n* = 145)		(*n* = 590)		
	*n* *		*n* *		
Age, years, Mnd (Q1 to Q3)	142	31 (28 to 36)	575	31 (28 to 35)	0.370
BMI before pregnancy **, kg/m^2^, Mnd (Q1 to Q3)	138	21.6 (19.9 to 24.0)	585	22.6 (20.7 to 25.6)	0.001
BMI group, *n* (%)					
<18.5		11 (8.0)		28 (4.8)	0.015
18.5–24.9		103 (74.6)		388 (66.3)	
25.0–29.9		20 (14.5)		114 (19.5)	
≥30		4 (2.9)		55 (9.4)	
Education level, *n* (%)	143		585		
Primary or unfinished secondary		3 (2.1)		55 (9.4)	<0.001
General secondary		13 (9.1)		129 (22.1)	
Vocational secondary		14 (9.8)		57 (9.7)	
Higher or unfinished higher		113 (79.0)		344 (58.8)	
Marital status, *n* (%)	144		590		
Married		110 (76.4)		376 (63.7)	0.014
Living in partnership		33 (22.9)		202 (34.2)	
Single		1 (0.7)		12 (20.1)	
Walking/cycling per day ***, *n* (%)	144		586		
<15 min		2 (1.4)		29 (4.9)	0.004
15–29 min		28 (19.4)		100 (17.1)	
30–60 min		72 (50.0)		215 (36.7)	
≥1 h		42 (29.2)		242 (41.3)	
Number of births, *n* (%)	95		588		
1		48 (50.6)		242 (41.2)	0.230
2		31 (32.6)		228 (38.8)	
≥3		16 (16.8)		118 (20.0)	
Time since previous delivery, years	49		340		
<1		2 (4.0)		4 (1.2)	0.268
1–3		16 (32.7)		102 (30.0)	
>3		31 (63.3)		234 (68.8)	

* available information. ** BMI (body mass index) According to the classification of the World Health Organization (WHO/UNICEF/UNO.IDA, 1998). *** during pregnancy.

**Table 2 nutrients-15-03493-t002:** Daily fintake of energy and macronutrients from food.

Variable	Pregnant Women (*n* = 145)	Postpartum (*n* = 573)	*p*
	Mnd (Q1 to Q3)	Mnd (Q1 to Q3)	
Energy, kcal	2338.5 (1788.9 to 2320.7)	2320.7 (1770.9 to 3013.9)	0.790
Protein, g	102.7 (78.6 to 142.6)	103.7 (81.2 to 136.4)	0.849
Carbohydrates, g	221.8 (168.6 to 280.5)	220.6 (166.2 to 296.5)	0.658
Fat, g	108.1 (77.6 to 137.2)	107.7 (80.4 to 142.4)	0.693

**Table 3 nutrients-15-03493-t003:** Daily dietary intake of vitamin D from various food groups and supplements.

Source	Pregnant Women (*n* = 145)		Postpartum (*n* = 573)		*p*
	Used (*n*, %)	Mnd (Q1 to Q3)	Used (*n*, %)	Mnd (Q1 to Q3)	
Food, mcg	145 (100.0)	2.5 (1.3 to 5.0)	573 (100.0)	2.0 (1.3 to 3.4)	0.006
Fish, mcg	132 (91.0)	1.33 (0.48 to 3.03)	518 (90.4)	0.85 (0.26 to 1.96)	<0.001
Dairy products, mcg	139 (98.6)	0.21 (0.10 to 0.40)	552 (96.3)	0.29 (0.16 to 0.42)	0.004
Eggs, mcg	143 (98.6)	0.61 (0.29 to 1.05)	566 (98.8	0.60 (0.30 to 0.94)	0.148
Mushrooms, mcg	129 (89.0)	0.04 (0.02 to 0.11)	448 (78.2)	0.02 (0.01 to 0.04)	<0.001
Butter, mcg	131 (90.3)	0.02 (0.01 to 0.05)	474 (82.7)	0.04 (0.01 to 0.05)	0.070
All supplements, mcg	136 (93.8)	62.5 (18.5 to 112.5)	480 (83.8)	35.0 (12.5 to 72.5)	0.107
Vitamin D, mcg	122 (84.1)	62.5 (25.0 to 112.5)	387 (67.5)	62.5 (25.0 to 112.5)	0.788
Multivitamins, mcg	76 (52.4)	7.8 (5.0 to 10.0)	282 (49.2)	10.0 (5.0 to 10.0)	0.011
Food and supplements, mcg	145 (100.0)	63.5 (20.2 to 104.4)	573 (100.0)	28.2 (11.1 to 73.9)	0.001

The “Used” column refers to the number and percentage of participants who consumed the determined food groups and supplements. The intake is calculated from this population.

**Table 4 nutrients-15-03493-t004:** Consumption of dietary vitamin D food groups.

Food g/Week	Pregnant Women (*n* = 145)		Postpartum (*n* = 573)		*p*
	Used (*n*, %)	Mnd (Q1 to Q3)	Used (*n*, %)	Mnd (Q1 to Q3)	
Fish	132 (91.0)	183.7 (93.2 to 344.6)	518 (90.4)	112.8 (50.6 to 237.2)	<0.001
Dairy products	139 (98.6)	350.4 (235.8 to 539.9)	552 (96.3)	358.0 (235.4 to 529.6)	0.682
Eggs	143 (98.6)	211.7 (99.0 to 368.2)	566 (98.8)	209.4 (103.6 to 329.1)	0.148
Butter	131 (90.3)	2.5 (1.1 to 5.0)	474 (82.7)	3.9 (1.1 to 5.0)	0.070
Mushrooms	129 (89.0)	28.8 (11.5 to 74.8)	448 (78.2)	11.5 (5.8 to 28.8)	<0.001

The “Used” column refers to the number and percentage of participants who consumed the determined food groups and supplements. The intake is calculated from this population.

**Table 5 nutrients-15-03493-t005:** Associations between vitamin D serum level and influencing factors.

Status	PTH Concentration, pmol/L	BMI, kg/m^2^	Education Level	Number of Births	Walking/ Cycling per Day	Sum of Vitamin D, mcg	Dietary Vitamin D, mcg	Supplement Vitamin D, mcg
Pregnant	−0.16	−0.06	−0.06	−0.28 *	−0.08	0.38 *	−0.07	0.31 *
Postpartum	−0.21 *	−0.14 *	0.20	−0.17 *	−0.06	0.36 *	0.02	0.15 *
ALL	−0.20 *	−0.17 *	0.19 *	−0.19 *	−0.08 *	0.38 *	0.03	0.18 *

* *p* < 0.05.

## Data Availability

To access the data used in this research, please contact the corresponding author.

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
