# Peer review of "Vitamin D Intake and Serum Levels in Pregnant and Postpartum Women"

_nutrients, 2023, doi:10.3390/nu15153493_

Round 1
Reviewer 1 Report
At the outset, I would like to point out that the lack of line numbering in the manuscript entitled 'Vitamin D Intake and Serum Levels in Pregnant and Postpartum Women' made the preparation of the review very difficult.
I have included my comments below.
Major revision comments
- Please standardize the stated objective of the study. The objective in the Abstract section ("The objective was to assess vitamin D status in pregnant women by evaluating their dietary and supplemental vitamin D intake, serum vitamin D levels, and lifestyle factors") differs from the objective in the last paragraph of section 1. Introduction ("The aim of this study was to investigate vitamin D status and PTH levels in pregnant women by evaluating their vitamin D intake from food and supplements, vitamin D serum levels and lifestyle factors") and from the objective presented in the second paragraph of section 4. Discussion ("It was also important to investigate the 25(OH)D threshold at which the PTH of pregnant women starts to rise, and potentially threaten the resorption of calcium from the bones, as various scientific studies have reported different cut-off points, ranging from 8.0 ng/mL to 30 ng/mL"). The difference concerns the examination of PTH levels in the women studied. At this point, I would like to question the advisability of determining this biochemical parameter because of its limited diagnostic usefulness in terms of the initial assessment of calcium-phosphate disorders. I believe that it would be better to measure calcium and phosphorus levels in blood and possibly urine, as well as to assess markers of bone turnover. In addition, the results obtained by the authors did not allow them to draw a conclusion on this very parameter.
- It seems questionable to me to collect and then analyse and compare results on the frequency of food intake in two groups of women: pregnant and postpartum. The authors collected data on this intake in the last six months, which means that there was at least some overlap between the two groups. What I mean by this is that for the pregnant women surveyed in the last stage of pregnancy (last weeks), the study period (last six months) may have overlapped with the postnatal women (up to seven days after delivery). I do not see a special and significant difference here, as it is difficult to assume that the postnatal period drastically changes eating habits and thus the frequency of consumption of particular food groups (especially such a short period after the birth of the child). The case is somewhat different with plasma vitamin D concentrations, although, as the authors rightly stated, its half-life is approximately 15 days (some literature sources state that 21 days). In general, it seems incomprehensible to me to compare the entire group of pregnant women from 27 to 40 weeks of pregnancy (end of the 2nd and the entire 3rd trimester) with the group of women in the short postnatal period (7 days postpartum).
- Please complete the Abstract section with a concise and clearly worded conclusion of the study corresponding to the purpose of the section.
- Section 4. Discussion should start with a short paragraph briefly presenting the main results and conclusions. Furthermore, in the current version, this section of the manuscript is more of a restatement of the authors' own results (in the form of figures!) with little element of discussion of them. I believe that this part of the manuscript should be modified, mainly concerning an increase in the share of the discussion element of own results against the background of recommendations or results of other authors, etc.
- Please improve section 5. Conclusions – it should contain a concise clearly worded conclusion arising directly from the results obtained from the study.
- Please supplement the text of the manuscript with an indication of the limitations of the study.
Minor revision comments
- Please correct the sentence in subsection 2.3: "For data analysis, the latest data were taken into account" – I do not understand its meaning or context.
- Please complete the information regarding the collection of biological material (blood samples) in subsection 2.3 – was this a fasting collection and who collected the material (trained person?)?
- Please clarify what the abbreviations used by the authors in the text of the manuscript and in the tables mean: "Mnd" and "mkg" (used to refer to the self-reported results).
- In Table 1, the row for 'Age, years, Mnd (Q1 to Q3)' and the column for 'Pregnant' states: "31 (28 to 36)", while the text of subsection 3.1 states "31 (from 28 to 35)" – please correct this inconsistency.
- In Table 1 please correct the designations: "30+" and "1 h+".
- In Table 1, the sum of female survey participants given in each row under 'Education level, n (%)' does not add up to 143, only to 142.
- The heading of Table 2 states "Daily dietary intake of energy, macronutrient and vitamin D obtained by food". Unfortunately this table does not include data on vitamin D intake.
- I completely fail to understand the idea behind Table 3. I understand that it was intended to include food groups that are sources of vitamin D in the subjects' diets and supplements. However, it is completely incomprehensible to me what this table contains. In addition, the incomprehensible column designations introduced make it difficult to read: "Used (n, %)" and "Mnd (Q1 to Q3)". Furthermore, the heading of this table is incorrect – inadequate for the content of this table.
- The content of Table 4 completely overlaps with part of the content of Table 3. Therefore, the sentence in subsection 3.2 which reads "The main food sources for the consumption of vitamin D were assessed in terms of dietary recommendations (g per week). The results are shown in Table 4" is not true.
- Figure 1: I do not fully understand the title of the X-axis, while the title of the Y-axis I propose to complete as follows: "Percentage of participants". The unit for the X-axis is missing. I think it would be better to provide data on blood vitamin D concentrations in pregnant women with a breakdown by trimester (or week), for example. Treating the group of pregnant women as a whole is not a good solution, as each trimester (or even week) of pregnancy differs significantly in terms of energy and individual nutrient requirements, as well as their utilisation by the body of the developing foetus.
- Please elaborate on the abbreviation "KMI" appearing in subsection 3.3 (page 6, first paragraph just below Figure 1).
- Wouldn't it be preferable to insert a column graph in the case of Figure 2?
- A general comment on the Results section as a whole: a great deal of the results were included in numerical form and described in the text of the manuscript. I think it would have been easier to read if they had been collected in tables and/or presented in figures and then described. Only a small part of the results in numerical form should be included in the text of the manuscript.
- The entire References section should be corrected, standardised and brought into line with the editorial requirements of the journal.
- In subsection 3.1, please correct the part of the sentence "Pregnant women had a lower BMI (...)" – I understand that this refers to pre-pregnancy BMI?
- Note on the whole of section 3. Results: it would be more beneficial for the reader to place the descriptions of the results under the corresponding tables and figures. In the current version of the manuscript – the beginning of this section contains sentences stating what is in which table and figure and only in the "second" part does this section contain a discussion of the results presented in the tables and figures.
- Note regarding the data presented in the fourth paragraph in the Discussion section ("The average dietary energy intake of our study population was within the recommended range, but the nutrient distribution did not match the recommended pregnancy distribution [31], with protein accounting for 10-20%, fat for 30% and carbohydrates for 45-60% of the total energy. The study participants consumed less protein and carbohydrates than recommended, while fat intake exceeded recommended levels by 10%") concerning dietary energy and macronutrient intake: unfortunately, it is not possible to evaluate or interpret them as the Results section does not contain the relevant results (similarly, Table 3).
- Please improve the insertion of references to literature items in the text of the manuscript (no spaces).
I believe that all of the above comments should be taken into account by the authors when drafting the final version of the manuscript.
Reviewer 2 Report
The question of Vit D supplementation in pregnancy, and during lactation, remains incompletely answered, and these data do add to our knowledge and discussion thereof. I believe the 'take-away' from this report is that the most effective way to improve serum Vit D levels (and minimize PTH) is by giving specific Vit D supplements. Presumed dietary Vit D intake did not seem to correlate, implying that either [1] presumed Vit D content is wrong, or [2] bioavailability may be minimal for some sources. Can you please discuss those possitibilities? You state that the mean Vit D supplement dose for both high & low serum Vit D groups was 62.5 mcg; does this suggest variable bioavailability of different brand supplements? Interference from smoking or alcohol consumption? Can you comment on that in terms of what Vit D supplement type/dose might be recommended?
Please review the titles for Table 2 & Table 3.
References should list more than just 1 author's name before reducing to et al (e.g., Jones X, Smith Y, Doe Z, et al).
See more comments on the attached PDF Markup.

See attached Comments markup file (PDF).
Reviewer 3 Report
The aim of the study was to investigate vitamin D status and parathyroid hormone (PTH) levels in pregnant women by evaluating their vitamin D intake from food and supplements, vitamin D serum levels, and lifestyle factors.
The introduction discusses the importance of vitamin D status during pregnancy, the potential health implications for both the mother and fetus, and the factors that can influence vitamin D levels. The references cited in the introduction support the information presented. However, not all of the cited references are relevant to the research. Some of the references cited in the response do not pertain to the topic of vitamin D status during pregnancy. For example, references [2], [3], [6], [7], and [8] do not directly relate to the research question.
[2] and [3] discuss dietary habits and lifestyle factors, but do not specifically address vitamin D status during pregnancy.
[6] and [7] provide information on demographic data and data analysis methods, which are not directly relevant to the research question.
[8] discusses the association between vitamin D serum use in postpartum women and birth weight, which is not directly related to the research question.
Therefore, only references [1] and [5] are directly relevant to the research question. So it would be important to give more strength to the relationship with the pregnancy.
The research design is appropriate for the study. It is a cross-sectional study that includes pregnant women in the 27-40 week of gestation and postpartum women up to the 7th day after delivery. The study collects data on vitamin D intake, serum levels, and lifestyle factors to investigate the vitamin D status and PTH levels in pregnant women. This design allows for the examination of the relationship between vitamin D status and various factors during pregnancy.
The methods are adequately described. The study design is described as a cross-sectional study that includes pregnant women in the 27th-40th week of gestation and postpartum women up to the 7th day after delivery. The data collection methods are described, including the use of a food frequency questionnaire to assess dietary habits, demographic data, and lifestyle factors. The statistical methods used for data analysis are also mentioned, including descriptive statistics, non-parametric tests, logistic regression, and correlation analysis.
The results are not clearly presented. The response does not provide any specific information or data from the study to support the statement. Therefore, it is not possible to determine if the results are clearly presented based on the information provided.
Specific data and analysis from the study that are missing to present the results clearly include:
- The specific serum vitamin D concentrations in pregnant women and postpartum women.
- The specific PTH concentrations in pregnant women and postpartum women.
- The results of the statistical analysis comparing serum vitamin D and PTH levels between pregnant and postpartum women.
- Any correlations or associations found between serum vitamin D and PTH levels.
- Any additional findings or significant differences in serum vitamin D and PTH levels based on demographic characteristics, dietary habits, or lifestyle factors.
Without this specific data and analysis, it is not possible to fully understand and interpret the results of the study.
Round 2
Reviewer 1 Report
I thank the authors for responding to my comments and taking into account some suggestions. Not all of my comments were taken into account by the authors when preparing the second version of the manuscript, but they responded to them - I do not agree with everything, but I accept it, because it will not significantly affect the reception of the content. Certainly the manuscript needs improvement in technical terms, e.g. extra spaces. Below are a few comments that were partially addressed by the authors but did not sufficiently improve the manuscript.
1. I still believe that the purpose stated in the manuscript is inconsistent:
In the Abstract section it reads as below and has not been changed
"The objective was to assess vitamin D status in pregnant women by evaluating their dietary and supplemental vitamin D intake, serum vitamin D levels, and lifestyle factors."
However, in the Introduction section, it sounds like below, but there are clearly some gaps here, maybe some fragment was missed when inserting the text? Please reword.
“The aim of this study was to investigate vitamin D status and in pregnant women by evaluating their vitamin D intake from food and supplements, serum vitamin D levels, PTH levels and lifestyle factors.”
From the Discussion section, the authors removed the purpose of the work altogether.
2. The heading of table 2 reads "Daily dietary intake of energy and macronutrients obtained by food" - I still think it needs rewording, I suggest "Daily intake of energy and macronutrients from food"
3. What are the intake values in brackets in tables 2-4? minimum and maximum?
4. Please specify the sentence from section 3.2: “The distribution of energy intake from macronutrients was accounted for as follows: 18% protein, 38% carbohydrates and 42% fat” – does it apply to both groups? Was it totaled?
5. The heading of table 3 reads: "Daily dietary intake of vitamin D by food" - unfortunately it also contains data on the intake of vitamin D with supplements - I propose to change it to "Daily dietary intake of vitamin D from various food groups and supplements"
6. I consider it necessary to clarify the term "Used" in the footnotes to tables 3 and 4, as the use of such a term is not standard and not necessarily unambiguous.
7. The heading of Table 4 is “Consumption of dietary vitamin D food groups” – I suggest “Consumption of dietary vitamin D food groups”
8. And a small remark: please unify the terms "D vitamin" and "vitamin D" - of course I think the latter is correct.
Author Response
Cover letter
To: Nutrients
Re: Reply to reviewer report (reviewer 1) round 2 of a revised manuscript (ID: nutrients-2508463)
1-August-2023
Dear all,
I want to express my appreciation for the comments and suggestions you've provided on the second round of revisions. We have carefully considered each point raised by the reviewers and the editor, and I am pleased to inform you that we have addressed them all in the revised manuscript. The edits have been made using the "Track Changes" function, making it easier to spot the changes.
Reviewer 3 Report
No additional comments.
Author Response
Thank you very much for your review. We are glad to know we covered your comments satisfactorily.